# Polymeric Antimicrobial Food Packaging and Its Applications

**DOI:** 10.3390/polym11030560

**Published:** 2019-03-25

**Authors:** Tianqi Huang, Yusheng Qian, Jia Wei, Chuncai Zhou

**Affiliations:** 1School of Materials Science and Engineering, Tongji University, Shanghai 201804, China; htq5699@163.com (T.H.); 1651601@tongji.edu.cn (Y.Q.); 2Department of Materials Science, Fudan University, Shanghai 200433, China

**Keywords:** antimicrobial agents, food packaging, biodegradable materials, antimicrobial packaging

## Abstract

Food corruption and spoilage caused by food-borne pathogens and microorganisms is a serious problem. As a result, the demand for antibacterial drugs in food packaging is growing. In this review, biodegradable and non-biodegradable materials for food packaging are discussed based on their properties. Most importantly, antibacterial agents are essential to inhibit the growth of bacteria in food. To keep food fresh and prolong the shelf life, different kinds of antibacterial agents were used. The composition and application of natural antibacterial agents and synthetic antibacterial agents are discussed. Compared with natural antibacterial agents, synthetic antibacterial agents have the advantages of low cost and high activity, but their toxicity is usually higher than that of natural antibacterial agents. Finally, future development of antimicrobial food packaging is proposed. It is an urgent problem for researchers to design and synthesize antibacterial drugs with high efficiency and low toxicity.

## 1. Introduction

Food decays when it is exposed to an open environment, such as during postprocessing, distribution, and storage. In daily life, food spoilage and deterioration are mainly caused by foodborne pathogens and other microorganisms, such as *Campylobacter, Salmonella, Yersinia enterocolitica, Escherichia coli,* and *Listeria monocytogenes*, which mainly grow on food surfaces; there are also some microorganisms growing in the volume of polymers throughout the pores and gels. [1,2]. Traditional food preservation methods include freezing, refrigeration, fermentation, drying, the use of additives (e.g., organic acids and salts), and thermal processing [3]. A relatively new concept called “active packaging” or “intelligent packaging” has been introduced [4] to provide interaction between food and packaging materials and to maintain a microenvironment. Active packaging extends the shelf life of food products [5]. Active packaging can provide several functions, such as scavenging of oxygen, moisture or ethylene; emission of ethanol and flavors; and antimicrobial activity [6]. Some of the most promising active packaging materials are antimicrobial films, which can be prepared by incorporating synthetic or natural antimicrobial agents into films or directly coating them on food [7]. However, antimicrobial coatings have several disadvantages. First, antimicrobial agents sprayed or dripped on the food cannot have an effective activity because of the rapid diffusion of antimicrobial agents and the reactions of these agents with the food [6]. Second, the addition of these agents negatively affects the taste of food. Antimicrobial agents carried by polymers interact with the surface to prevent foodborne microbial growth and control the diffusion rate of antimicrobial agents, thereby ensuring the continuous and adequate presence of antimicrobials on food surface [8]. Antimicrobial food packaging has a slight negative influence on food, and provide excellent antimicrobial properties that can preserve food for a long time [9]. Therefore, antimicrobial agents are often incorporated into films to develop antimicrobial food packaging.

The antimicrobial abilities of food packaging depend on antimicrobial agents, which can be classified as natural antibacterial agents and synthetic antibacterial agents based on their sources. Antimicrobial agents have been developed from natural sources because natural antimicrobial agents are relatively safe and are easy to obtain [10]. Natural antimicrobial agents are extracted from materials in nature and require sophisticated extraction procedures. The amount of natural antimicrobial agents depends on the kinds of original animals or plants, which may be rare or precious. However, with increasing food production and demand for packaging, natural antimicrobial materials tend to be insufficient. Thus, organic or inorganic synthetic antimicrobial agents have been developed. The main antibacterial components in food packaging are synthetic organic materials, which mainly contain ethylene diamine tetraacetic acid (EDTA), fungicides, parabens and other chemicals.

Polymeric food packaging films are some of the most commonly used films because they are easy to produce and have excellent performance [11]. Polymeric food packaging can protect food from microbial attack and has properties such as flexibility, strength, stiffness and a barrier to oxygen and moisture. This review introduces polymeric film-based antibacterial packaging and focuses on natural antimicrobial packaging and synthetic organic antimicrobial packaging, according to the source of antimicrobial agents. Table 1 shows some examples of incorporation of antimicrobial agent into polymeric matrix and their applications in food packaging. Other studies on the microbiological and physical properties of antibacterial packaging materials are discussed according to the types of antimicrobial agents in the following text.

## 2. Polymeric Matrix Used in Antimicrobial Packaging

The reported polymeric food packaging materials can be further divided into nonbiodegradable and biodegradable polymer antimicrobial food packaging based on the property of polymeric matrix.

Nonbiodegradable polymer used in the antimicrobial food packaging mainly include poly(vinyl chloride) (PVC), polypropylene (PP), high density polyethylene (HDPE), low density polyethylene (LDPE), poly(ethylene-co-vinylacetate) (EVA), poly(ethylene terephthalate) (PET) and so on. Plastics are commonly used as food packaging, and their use is continuously increasing because of the inherent plastic film properties, such as ease of processing, transparency and flexibility. 

**Polyvinyl chloride (PVC)** is the most widely-used plastic product in the world; it is a common food packaging material and can be easily utilized to produce materials with antimicrobial property. PVC films have several advantages, including flexibility, softness, toughness, medium strength, chemical inertness, high permeability in inherent performance, ease of heat sealing, and excellent self-cling properties in processing [27]. They are normally mixtures of PVC resin, plasticizers and other additives, and they exist in oriented and unoriented films.

**Polypropylene (PP)** is a thermoplastic resin prepared via propylene polymerization. In comparison with other commodity plastics, the density of PP is low (0.89–0.91 g/cm^3^) [27]. PP can effectively resist water vapor. It also has a high melting point, making it suitable for high temperature packing applications.

**High density polyethylene (HDPE)** is a linear polymer of ethylene with only a small branching. HDPE is stiffer but less transparent than LDPE. HDPE films have a good gas and water vapor barrier, but are highly permeable to oxygen and carbon dioxide. As food packaging, HDPE films are durable and can preserve food for a long time.

**Low density polyethylene (LDPE)** as another commonly used polymeric packaging material, is the cheapest among plastic films. As food packaging, LDPE can effectively resist water vapor with good transparency. However, LDPE cannot hold back oxygen, carbon dioxide and some other vapor compounds. LDPE can be used as a film for heat sealing [28].

**Poly(ethylene-co-vinylacetate) (EVA)** films are commonly used as food packaging in refrigerated items. EVA is composed of copolymerized PE and PVA, which has a low vapor permeability [29]. EVA films are widely used in food packaging applications because of their flexibility, heat sealability, high adhesivity, nontoxicity, good clarity and stretch properties [30,31]. With these characteristics, EVA and EVA based composites have been widely explored.

**Poly(ethylene terephthalate) (PET)** films are common, useful packaging materials for potentially replacing PVC. They have the advantages of a relatively good mechanical strength, toughness, and a high crystalline melting temperature (270 °C). They are mainly used as bottles and food packages due to their safety and easy processing. The demand for PET with various shapes and without residual acetaldehyde is increasing [32].

Although food packages composed of nondegradable plastics remain the most commonly used materials, their use is decreasing because they cause environmental pollution. In contrast, biodegradable food packaging is more promising in the long run because it is green, reproducible, and environmentally friendly. In countries where landfill is the main method of waste management, the use of biodegradable materials is the most promising with the development of biodegradable polymer antimicrobial packaging, such as polylactic acid (PLA), cellulose, starch, and chitosan.

**Polylactic acid (PLA)** is considered as one of the most promising and environmentally-friendly polymers because of its excellent physical and chemical properties, including renewability, biodegradability, and biocompatibility [33]. PLA derived from corn is biodegradable, which makes it unique in food applications. PLA has generally considered as safe (GRAS) status, as given by the USFDA (US Food and Drug Administration). However, one of the major disadvantages of PLA in food packaging is its high permeability to gas and vapor, limiting its use for short-life packaged food [34]. Given that PLA is nontoxic, noncarcinogenic, biocompatible, hydrophilicity, water soluble and chemically stable, it is usually mixed with different polymers.

**Cellulose** is the most abundant natural polymer, which can be extracted from corncobs. It is environmentally friendly and biodegradable. The most used raw material, which is livestock cellulose, is a linear homopolysaccharide [35]. Composite films made of cellulose have attracted much attention because of their recyclability and degradability. However, their high cost and sensitivity to water limit their applications.

**Starch** can be used as a food packaging material and often acts as a thickener, adhesive or additive. Starch has the advantage of being a moderate oil barrier, but it is weak in a moist environment because of the hydrophilic function groups in its molecular structure. Its crystalline region can form a parclose against outside gases, and the effect becomes stronger for amylose films than for amylopectin ones [36].

**Chitosan** is produced from chitin with a linear structure, which is constituted by random deacetylated unit and acetylated unit. Chitosan is an antimicrobial polymer and can be used as an antimicrobial agent and polymer substrate at the same time. Chitosan films used as environmentally friendly food packaging prolong the shelf life and preserve fresh food due to their good antimicrobial properties [37]. It is a weak base so that it can dissolve in acidic solutions, but it remains insoluble in water. It can be produced by extrusion and press molding techniques, making it easy to exhibit antimicrobial properties in the form of films. 

## 3. Natural Antimicrobial Agent and Their Application in Food Packaging

### 3.1. Essential Oils (EOs)

EOs are complex lipid oils naturally produced as secondary metabolites by plant materials, such as flowers, buds, seeds, leaves, wood, fruits, roots, twigs, and barks. EOs are mainly mixtures of natural volatile compounds obtained via compression or distillation from plants [38,39]. EOs contain various compounds, such as monoterpenes, diterpenes, and triterpenes, including their oxygenated derivatives, and phenolic compounds, which are responsible for the antibacterial properties of EOs (Figure 1) [40]. In addition to their antibacterial action, EOs also possess antifungal, antioxidant, antiviral, antimycotic, antitoxigenic, antiparasitic, antibiotic, and antiseptic properties. According to the Food and Drug Administration, EOs have Generally Recognized as Safe (GRAS) status [41]. Given their excellent biological performance, EOs are the most widely used additives in the food industry.

EOs have been used in food packaging to extend shelf life. Three methods are employed to utilize EOs as antimicrobial agents: mixing EOs into basic materials, coating EOs onto food packages and loading EOs into an antibacterial pouch. Among these methods, mixing EOs with other polymers is commonly used because it is convenient and easily applied to mass production. Nowadays, environmental friendliness is the common trend of scientific research. Under this circumstance, EOs incorporated with biodegradable polymers, such as PLA, skate skin gelatin, and chitosan, are popular throughout the world.

Misaghi et al. [42] incorporated different concentrations of *Mentha piperita* EO (MPO), *Bunium percicum* EO (BPO), and nanocellulose (NC) into PLA films to produce antimicrobial films. They stored ground beef sealed with well-prepared films at 4 °C for 12 days and assessed organoleptic and antimicrobial effects. They found that MPO and BPO have an antimicrobial activity against *Staphylococcus aureus*, *Enterobacteriaceae*, and *Pseudomonas*. The film prepared with a solution of 1% *w*/*v* PLA, 0.5% *v*/*v* MPO, and 1% *v*/*v* NC has the best effect in terms of antibacterial and organoleptic characteristics. With EO addition, the performance of water vapor barrier is improved, but some morphological parameters were compromised. Copaiba oil with anti-*bacillus* subtilis properties is embedded into paper and PLA film. Both films show antimicrobial activity against *B. subtilis* when the EO content reaches 20 wt % [43].

Llana-Ruiz-Cabello et al. [44] introduced different concentrations of Origanum vulgare L. virens essential oil (OEO) into PLA films and evaluated the antimicrobial activity of the resulting film for use in ready-to-eat salads. This new active packaging shows antimicrobial properties against yeasts and molds, and 5% and 10% of OEO are the most effective. The resulting film as an active food packaging exhibits suitable mechanical and physical properties with slight changes. Jouki et al. [45] investigated the physical, thermal, barrier, antioxidant and antibacterial properties of Quince seed mucilage (QSM) films after OEO is incorporated. The film with 1% OEO has no effect on Salmonella typhimurium and Pseudomonas aeruginosa, but strongly inhibits the growth of *S. aureus* and *E. coli*.

Sweet potato Starch/montmorillonite (MMT) nanoclay nanocomposite films activated with thyme essential oil (TEO) are developed as an active biodegradable nanocomposite film for food packaging [46]. MMT is added to improve the mechanical property of starch films. The results show that *E. coli* and *S. Typhi* (p < 0.05) volumes decrease to the detectable levels after EOs are incorporated within 5 days, whereas the control group without EOs has approximately 4.5 log colony forming unit (CFU)/g. Song et al. [47] incorporated TEO into a biodegradable skate skin gelatin (SSG) film as an antimicrobial agent. They found that the SSG film with 1% of TEO in the packaging of chicken tenderloin samples effectively inhibits the growth of *L. monocytogenes* and *E. coli O157:H7*. 

Rezaei et al. [48] incorporated rosemary essential oil (REO) into a chitosan film, which is often used in food packaging because of its antimicrobial and aspirating properties. The antibacterial properties of a chitosan film with REO are better than those of pure chitosan film.

EOs also combine with nonbiodegradable polymers for easy production and processing. However, traditional processing methods have many limitations, resulting in the development of many new processing methods. Electrospinning is one of the most popular methods to prepare nanofibers. With an external electric field, polymer solutions or melting polymers can form nanofibers with diameters in the sub-micron range. This method has several advantages, such as high surface area and highly porous structure.

Cui et al. [49,50] utilized electrospinning technology to produce antimicrobial food packages. Utilizing the advantages of electrospinning, they encapsulated tea tree oil (TTO) and beta-cyclodextrin (β-CD) into poly(ethylene oxide) (PEO) by electrospinning. After administering plasma treatment, they further determined the TTO antibacterial activity against *E. coli* O157:H7 on beef at different temperatures for 7 days. They found that the antibacterial properties greatly increased after the plasma treatment, and an inhibition efficiency of 99.99% was obtained at 4 or 12 °C, suggesting that TTO can prolong the shelf life on beef. By incorporating cinnamon essential oil/β-cyclodextrin (CEO/β-CD) proteoliposomes into PEO nanofibers by electrospinning technique, the controlled release of CEO from proteoliposomes was achieved via proteolysis of protein in proteoliposomes (Figure 2). These studies demonstrated that antimicrobial properties can be improved via nanofiber encapsulation.

### 3.2. Bacteriocins

Bacteriocins are bactericidal substances produced by bacteria, encoded by genes, and synthesized by ribosomes. Bacteriocins come from a wide range of sources and have various structures, which provide the possibility for bacteriocins development as new kind of antimicrobial agents. Nisin and pediocin are commonly used bacteriocins.

Nisin, an antibacterial peptide produced by a lactic acid bacterium and *Lactococcus lactics*, is the most widely used bacteriocin in active food packaging. Nisin is used in more than 48 countries and approved by the USFDA. Nisin can inhibit the growth of a broad spectrum of Gram-positive microorganisms, such as *Listeria*
*monocytogenes* [51,52] and prevent spore germination [53]. Nisin has both positive charges and hydrophobic groups. The positive charge causes electrostatic interaction between peptide and bacteria’s negatively charged cell membranes, allowing the antimicrobial peptide to attach to the bacterial cell membrane [54]. Given that the bacterial cell membrane is also hydrophobic, the hydrophobic amino acid of antibacterial peptide can insert into bacterial cell, thereby changing the permeability of the bacterial cell membrane. Cellular substances, such as DNA, leak out of cells, causing bacterial death. Nisin can be easily combined with polymers to exhibit antimicrobial properties by coating on the polymer films or mixing with polymers and then producing films as food packages. Numerous studies have reported the effectiveness of insin-containing active packaging in preventing the growth of different food-borne bacteria.

Sugandha Bhatia et al. [55] evaluated the antibacterial activities of nisin, EDTA and lysozyme incorporated in a starch film-based food packaging film. They confirmed the effectiveness of the combined effect of antibacterial agents through antibacterial experiments, supporting the hypothesis that nisin and EDTA have a partial synergistic effect on antibacterial activities. Moradi et al. [56] fabricated a composite film containing chitosan, cellulose, and nisin and used it as an antimicrobial package of ultra-filtered (UF) cheese. The pure chitosan-cellulose film does not exhibit antimicrobial activities against *L. monocytogenes*, whereas a film incorporated with nisin shows a significant increase in the inhibitory effects on the *L. monocytogenes*, further demonstrating the antimicrobial properties of nisin. UF white cheese stored at 4 °C for 14 days packaged with a composite nisin film does not significantly change upon exposure to *L. monocytogenes*.

Nisin can inhibit the growth of microbes and keep food fresh even at room temperature. For example, nisin has antimicrobial properties on the seasoned beef, which is a popular food in Korea [57]. Seasoned beef is packaged and kept in storage for 60 days at low temperature (4 °C) and room temperature (25 °C). As a result, seasoned beef with nisin favors the slight increase in the growth of mesophilic microorganisms, whereas seasoned beef without nisin has a remarkable increase in the growth of these microorganisms. That means nisin has excellent antimicrobial properties, even at a room temperature. Nisin improves the mechanical properties of food packages, thereby contributing to the preservation of seasoned beef. Packages containing nisin slightly change in cutting force during storage at 25 °C, whereas packages without nisin show a remarkable decrease. Therefore, nisin is a promising antimicrobial agent widely used in food package.

In addition to the antimicrobial property of nisin, its release rates are essential for food preservation. Imran et al. [58] studied the release rates of nisin from hydroxypropyl methylcellulose (HPMC), chitosan, sodium caseinate, and PLA films at the temperature of 4 and 40 °C in a water-ethanol solution. They found that HPMC has the largest nisin diffusion coefficient, whereas chitosan has the smallest diffusion coefficient. Temperature was proportional to the release rate.

Dykes et al. [59] prepared a cellulose film containing nisin for meat packaging applications. *L. monocytogenes* displays a decrease in growth after 14 days of storage, indicating that the antimicrobial properties of food packaging are enhanced. They demonstrated that nisin can be added to cellulose films to fabricate a green and high-performance film. Bras et al. [60] developed green and convenient cellulose fiber-based nisin used for antimicrobial food packaging. They studied the two different methods that can introduce nisin to food packages: mixing nisin with basic food packages and grafting nisin to basic food packages. Nisin-grafted carboxylated cellulose nanofibers show antimicrobial properties against various Gram-positive bacteria. Nisin mixing with cellulose nanofibers can kill *B. subtilis*. Mixing nisin with cellulose naofibers is a more convenient method than grafting, but the former retains antimicrobial activities for a shorter time than the latter.

Correa et al. [61] developed a biodegradable polyhydroxybutyrate /polycaprolactone film with nisin. The film can effectively inhibit *L plantarum* CRL691 existing on the sliced ham and has potential for applications in processed meat packaging. Marcos et al. [62] utilized antimicrobial packaging with nisin-based polyvinyl alcohol to reduce the levels of *L. monocytogenes* levels of sliced fermented sausages without added sodium salt. They found that antimicrobial properties are affected by the processing treatment and provided a new guide for antimicrobial food packaging. 

Pediocin is another commonly used bacteriocin, which is mainly derived from *pediococcus acidilactici*. It is an unmodified small-molecule protein with good thermal stability, stable physicochemical properties, and broad antimicrobial spectrum, pediocin has a wide range of adaptability to temperature and pH. It can be degraded in the human body because of the nature of polypeptide, and can effectively inhibit various foodborne pathogenic bacteria. Thus, pediocin can be developed as a new generation of natural food preservatives [63,64].

Soares et al. [65] developed a cellulose matrix film with a high antimicrobial efficiency based on pediocin, which can inhibit the growth of the pathogenic bacteria. They stored ham slices for up to 15 days and found that films with 25% and 50% pediocin have a similar performance in terms of antimicrobial properties, that is, the films cause a 0.5 log cycle reduction after the ham slices are stored for 12 days. They also fabricated nanocomposite films containing pediocin and ZnO nanoparticles with antimicrobial properties against *L. monocytogens* and *S. aureus* [66]. The addition of pediocin results in increased values of elongation at break. The presence of pediocin in nanocomposite films produces slightly yellowish films, which are balanced by ZnO nanoparticles incorporation, resulting in a whitish coloration. 

Ramana et al. [67] incorporated crude pediocin into polyhydroxybutyrate (PHB) to develop antimicrobial films. They found that the film shows antimicrobial activity against foodborne pathogens, spoilage bacteria, and fungi. However, crude pediocin is unable to inhibit fungal, further demonstrating that the combined effect of pediocin and PHB films favor antifungal activity.

Although nisin is the only natural preservative for bacteriocins approved by the joint food additives committee (FAO/WHO), the safety of pediocin has been an indubitable fact. Pediocin is likely the most commercialized natural preservative after nisin.

### 3.3. Lysozyme

Lysozyme, an important class of enzyme, is a hydrophilic monopeptide chain [68]. It can inhibit bacterial infections, especially those caused by Gram-positive bacteria. The antimicrobial activity of lysozyme is due to its ability to hydrolyze the beta-1–4 glycosidic bonds between *N*-acetylmuramic acid and *N*-acetylglucosamine in peptidoglycans [69]. Lysozyme hydrolyzes peptidoglycan, which is the main cell wall component, destroying the cell wall, causing intracellular materials to leak out and resulting in bacterial death [70].

Numerous studies have been performed to develop food packaging materials based on antimicrobial enzymes immobilization by physical blending or chemical bonding. For example, an antimicrobial active packaging based on lysozyme is prepared via the sol–gel route by using PET as a matrix to develop packaging films with controlled release properties [71]. The film shows an antimicrobial activity against *Micrococcus lysodeikticus*, suggesting that lysozyme possibly migrates from the film to water and remains active after it is incorporated into PET films. 

Yemenicioglu et al. [72] incorporated partially purified lysozyme into zein films. The release rate of lysozyme under 4 °C changes between 7 and 29 U/cm^2^·min and increased at high lysozyme concentrations. They also tested the antimicrobial activity of the films against *B. subtilis* and *Lactobacillus plantarum*. The activation of partially purified lysozyme in zein films has the advantage of obtaining an antimicrobial effect in zein films with low initial lysozyme activities. However, the low solubility of partially purified lysozyme causes the formation of protein aggregates in zein films. This can be improved by the addition of edible ingredients with an emulsifying activity. Altinkaya et al. [73] attempted to modify the release rate of lysozyme in cellulose acetate films by changing the structure of films from highly asymmetric and porous to compact. The structure of the film surface influences the lysozyme release rate because the main release mechanism is the diffusion of lysozyme through films. This phenomenon is significantly influenced by the morphological characteristics of the films. This work shows that asymmetric porous films containing antimicrobial agents can be used as novel internal food packaging materials with controlled release properties. However, further studies are needed to investigate the effectiveness of these films on selected food systems.

Goddard et al. [68] developed a novel antimicrobial film that covalently attaches lysozyme to two different ethylene vinyl alcohol copolymers (i.e., EVOH 29 and EVOH 44). The antibacterial activities of the resulting films against *L. monocytogenes* are close to those of the free enzyme. The log reduction values of EVOH 29−lysozyme, EVOH 44−lysozyme, and free lysozyme are 1.08, 0.95, and 1.34, respectively. Del Nobile et al. [74] incorporated different concentrations of lysozyme into a plasma-treated PE film through chemical immobilization. Plasma treatment process involves the physical or chemical modification of a surface layer of the PE film, thereby contributing to the later immobilization of lysozyme on the surface. They found that plasma-treated films containing lysozyme have an antimicrobial performance against *M. lysodeikticus*.

However, in some cases, such as Ugi reaction with cyclohexyl isocyanide and glutaraldehyde to polyamide, lysozyme immobilization via covalent attachment seems unsuitable because of the substantial loss of enzyme activity [75].

### 3.4. Organic Acid

Organic acids, such as propionic acid, lactic acid, malic acid, sorbic acid, and tartaric acid, are commonly used as traditional food additives. Sodium benzoate and potassium sorbate have a GRAS status [76], and they can inhibit the growth of bacterial and fungal cells [70]. Sodium citrate inhibits the growth and reproduction of Listeria, bladder, and Escherichia coli O157:H7 [77]. Sorbic acid and potassium sorbate show activity against a broad spectrum of bacteria and molds [76]. At pH from 6.0–6.5, the preservative effect of sorbic acid is 5–10 times more resistant than that of benzoate [78]. Nano-scale benzoic acid and sorbic acid have better antibacterial properties than their non-nano-scale equivalent [9]. As such, a small amount of nano-sized antimicrobial agents used in antimicrobial packaging can obtain acceptable antibacterial effects. 

With the use of these organic acids and organic acidic salts in food packaging to extend shelf life, the antibacterial mechanism of organic acids has been studied. Organic acids have two forms: dissociated and undissociated. In a solution, the amount of organic acids undergoing dissociation and undissociation is in a dynamic equilibrium state. The unionized form of organic acids can cross the cell membrane and penetrate cells, and dissociate in cells to produce protons because of the high pH in cells. Bacterial cells consume energy to transport protons out of their cell membranes to keep the pH inside them at around 7. The transport of these protons exhausts all their energy in bacterial cells [79]. 

Numerous examples show that bacterial growth can be inhibited by organic acid-based food packaging and extend the shelf life of foods. Liu et al. [80] developed antimicrobial EVOH films with sorbic acid-chitosan microcapsules (S-MPs) and applied them to fish fillets. In antimicrobial tests, the film inhibits the growth of *E. coli*, *S. Enteritidis* and *L. monocytogenes* and prolongs the shelf life of fish fillets. In comparison with the addition of antibacterial agents directly to film, the incorporation of antibacterial agents by using the chitosan microcapsule can reduce the release rate of sorbic acid in films. Microcapsules incorporated with antibacterial and antioxidant active agents may offer a broad application prospect as food packaging materials. 

Starch-based films are a common and economical antimicrobial packaging materials. Lopez et al. [81] incorporated potassium sorbate into different formulations of starch films at different pH levels. They found that starch type and pH change do not affect the kinetic release. The film can inhibit the growth of *Candida* spp., *Penicillium* spp., *S. aureus and Salmonella* spp. Applying this film to frozen cheese can extend its shelf life by 21%. Salleh and Muhamad [82] fabricated antimicrobial packaging based on wheat starch containing lauric acid and chitosan, which act as the antimicrobial agents. They tested *B. substilis* and *E. coli* to confirm the antimicrobial property of the multicomponent film and showed that the film more effectively inhibited *B. substilis* than *E. coli* in solid media or liquid culture, indicating that the multicomponent film used for food packaging has a better effect on *B. substilis*. Therefore, it can be used to protect fresh food from bacterial attack.

### 3.5. Chitosan

Chitosan is a green and biocompatible material that has intrinsic antimicrobial properties and is widely used in the fields of biomedicine, food packaging, environmental conservation [3,83]. High-molecular weight chitosan has an antimicrobial activity, with mean minimum inhibitory concentration (MIC) values of 0.010% and 0.015% *w*/*v*, respectively [9]. MIC is the minimum concentration of drug that can inhibit the growth of certain microorganism, which is used for the quantitative determination of antibacterial activity in vitro. In terms of film forming ability, antibacterial ability, and biodegradability, chitosan is a good candidate for antibacterial packaging.

The antimicrobial mechanism of chitosan has been proposed to be mainly divided into three types. (I) Electrostatic attractions exist between positively charged chitosan chains and negatively charged bacterial cell walls, causing the absorption of chitosan onto the target bacteria. Thus, intracellular components tend to break down, leading to cell death. (II) The accumulation of chitosan can envelop the target bacteria, isolating the bacteria from the environment and inhibiting nutrient intake and exchange. (III) Chitosan has a chelating effect on metals and oligoelements, which are involved in antibacterial activity. Metals and oligoelements are essential for bacterial growth, and their absence results in bacterial death. The drawback of chitosan is its poor processing property, leading difficulty for chitosan to be processed into food packaging.

As a biopolymer, chitosan has a good film forming ability and is easily used with other bioactive agents, giving chitosan good comprehensive properties. Rezaei et al. [84] mixed two antimicrobial agents based on chitosan and cinnamon EO to fabricate a high-performance biodegradable film. The film not only shows increased antimicrobial activity, but also exhibits low affinity to water. These phenomena are the result of the combined effect between cinnamon EO components and the chitosan matrix. Chawla et al. [85] prepared composite films from chitosan and polyvinyl alcohol containing mint extract/pomegranate peel extract. The addition of extracts improves the tensile strength of the films by up to 41.07 ± 0.88 MPa without affecting their puncture strength. The films also have good antioxidant properties provided by the extracts and antibacterial activity against *S. aureus* and *Bacillus cereus*. Siripatrawan et al. [86] incorporated green tea extract into a chitosan film to develop active packaging for pork sausages. The film successfully inhibits the microbial growth at 4 °C, showing enhanced antioxidant and antimicrobial properties and providing food protection. Chitosan films with 0, 2.5%, 5%, 10% and 20% *w*/*w* propolis extract have also been developed. Polyphenols are the main molecule in the propolis extract. The mechanical property and antioxidant activity of the film are greatly enhanced. They proposed that some interactions occur between chitosan and propolis extract, thereby increasing the antimicrobial properties of the films.

However, chitosan films have several disadvantages, including rapid dissolution in acidic solutions and poor mechanical properties as food packaging, especially in a wet environment. Li et al. [87] fabricated a biodegradable composite film containing silica, PVA and chitosan (Figure 3). The film is biodegradable, low cost, and high performing and can be used in food packaging. With 0.6 wt % SiO_2_, the tensile strength of the PVA/ chitosan films is as high as 44.12 MPa and improved by 45% through hydrogen bonds between silica and PVA or chitosan. SiO_2_ also reduces the moisture and oxygen permeability of the food packaging films to retain freshness. The weight of PVA/chitosan decreases to 60% after 30 days in soil, suggesting its degradability compared with other nondegradable plastic food packages. The composite film is useful and had a promising for the future green, high-performance and environmentally friendly food packaging.

The electrospinning of chitosan in the form of nanofibers is another promising process that has been widely explored. Ajji et al. [88] fabricated chitosan-based nanofibers via electrospinning to prevent the spoilage of meat and the growth of pathogenic bacteria, such as *E. coli, S. enterica serovar Typhimurium, S. aureus and L. innocua.* They demonstrated that antimicrobial property of chitosan-based fibers is based on the protonation of amino groups, and this observation is consistent with mechanism (I) of chitosan. They also found the following susceptibility order: *E.coli > L.innocua > S. aureus > S. Tyohimurium*, which was a guide for meat quality preservation.

### 3.6. Grape Fruit Seed Extract (GFSE)

GFSE contains abundant phenolic compounds such as catechins, epicatechin, gallic acid and procyanidins. Therefore, GFSE is a naturally derived antimicrobial agent known to display a wide range of microbial growth inhibition against both Gram-positive and Gram-negative bacteria [89,90]. GFSE possesses strong antiseptic, germicidal, antibacterial, fungicidal and antiviral properties.

Thian et al. [91,92] incorporated GFSE into poly-ε-caprolactone, chitosan and polyethylene to fabricate antibacterial food packaging with promising properties. The incorporation of GFSE to biodegradable poly-ε-caprolactone increases the crystallinity of poly-ε-caprolactone and alginic acid [92]. The smooth and homogeneous film is made via compression molding. The film also has an excellent antimicrobial activity against *P. aeruginosa*. GFSE can be well dispersed into chitosan at different amounts (0.5%, 1.0% and 1.5% *v*/*v*) without affecting film transparency. Increasing GFSE induces the tensile strength of chitosan-based composite films and improves the elongation at break of the films. Preliminary antifungal activity evaluation has demonstrated that chitosan-based composite films can retard fungal growth. They incorporated GFSE into a polyethylene film via co-extrusion or solution-coating process and tested the antimicrobial property of this film by applying it to ground beef at 3 °C. The composite film shows an enhanced antimicrobial activity against several microorganisms, including *E. coli* IFO 3301, *S. aureus* IFO 3060 and *B. subtilis* IFO 12113. 

Park et al. [93] developed multicomponent antimicrobial agents, including lysozyme, nisin, GFSE, and EDTA, which work alone and in combination. They found that GFSE-EDTA is a widely applicable antimicrobial agent against all indicator microorganisms, especially Gram-negative bacteria, in Na-alginate- and K-carrageenan-based films.

### 3.7. Allyl Isothiocyanate

Allyl isothiocyanate (AIT), which has a strong antibacterial activity against many microorganisms, is the main flavoring ingredient found in the seeds, stem, leaves, and roots of cruciferous plants, such as mustard, wasabi, and horseradish; AIT has been produced via chemical synthesis due to mass market demand [70,94,95]., AIT has a GRAS status in the United States and is permitted for use as a food additive in Japan [96,97].

As a volatile antimicrobial agent, AIT is expected to be gradually released into the atmosphere from packaging materials, thereby inhibiting the growth of undesirable microorganisms on food surfaces. For this purpose, LDPE capsules containing AIT beads have been developed [98]. The effects of film thickness and temperature on release rate of AIT have been studied. As LDPE thickness decreases and temperature increases, the release of AIT increases obviously. The antibacterial test shows that the film inhibits the growth of *E. coli O157:H7*, molds, and yeasts on fresh spinach. The initial count of *E. coli O157:H7* is 5.6 log CFU/leaf decreases by 1.6–2.6 log CFU/leaf at 4 °C within 5 days.

## 4. Synthetic Organic Antimicrobial Agent and Their Application in Food Packaging

### 4.1. EDTA

EDTA, a chelator, can have an antimicrobial effect due to limited cation availability and the ability of EDTA to destabilize bacterial cell membranes via the complexion of divalent cations, which act as salt bridges between membrane macromolecules, such as lipopolysaccharides [99,100]. 

The synergism of EDTA with other antimicrobial agents has been widely studied. EDTA can enhance the antimicrobial activity of weak acids against Gram-negative bacteria [101]. Economou et al. [99] demonstrated that EDTA can help enhance the antibacterial effect of nisin and extend the shelf life of food. The combined effect of EDTA with an oregano EO on modified atmosphere-packaged chicken liver meat has also been investigated [102]. Microbiological data have shown that the combined use of EDTA (20 mM) oregano EO (0.1% and0.3% v/wt) under modified atmosphere packaging (30%CO_2_/70%N_2_) can extend the shelf life in comparison with aerobic packaging.

EDTA can improve the sensitivity of Gram-positive bacteria and Gram-negative bacteria to lysozyme [70,101]. Del Nobile et al. [103] studied the antimicrobial effect of nisin and lysozyme combined with EDTA on spoilage microorganisms in chilled buffalo meat. The best effect is achieved by combining 0.5% lysozyme and 2% EDTA, which can inhibit the growth of all investigated bacteria, including *Brochotrix thermosphacta*. These studies have confirmed that incorporating EDTA into some antimicrobial agents can enhance their inhibitory effects on spoilage bacteria. 

### 4.2. Fungicides

In food packaging, imazalil is used as an antimycotic agent. Imazalil(1-[2-(2,4-dichlorophenyl)-2-(2-propenyloxy)ethyl]-l*H*-imidazole) is a fungal sterol biosynthesis inhibitor, which is commonly used to control a wide range of fungi [70].

Hotchkiss et al. [104] investigated the general mechanism by which an antimycotic-incorporated LDPE film can inhibit surface mold growth in packaged foods. The LDPE film containing concentration of 2000 mg/kg imazalil can delay the growth of *A. toxicarius* on potato dextrose agar, whereas the LDPE film containing 1000 mg/kg imazalil markedly delays the growth of *Penicillium* sp. and mold on cheddar cheese. Thus, the incorporation of an antimycotic agent, such as imazalil, into food contact packaging films can inhibit surface mold growth.

Vartiainen et al. [105] prepared a food packaging film by incorporating imazalil into 2 mm-thick LDPE films. The molded plates containing 0.05–0.25% has a strong antimicrobial activity against *A. niger*. Imazalil can also highly inhibit *A. niger* growth with 30 mm inhibition zones. Cohen et al. [106] studied the activity of imazalil incorporated into HDPE film by using four different citrus varieties. They found that imazalil significantly reduces the degree of putrefaction in all varieties of citrus fruits.

These examples confirm that incorporating imazalil into food packaging film can effectively inhibit mold growth. However, pesticide overuse has led to the development of fungicide-resistant strains, and concerns about the safety of humans have led to the search for other antimicrobials [107].

### 4.3. Propyl Paraben

Propyl paraben is a colorless and odorless substance with a GRAS status [26]. Propyl paraben can inhibit the growth of molds and yeasts. At pH 4–8, it has a strong antifungal activity [108]. The antibacterial action mechanism of propyl paraben may be related to its effect on cytoplasmic membrane. Yam et al. [26] studied the antimicrobial activity of propyl paraben against *Saccharomyces cerevisiae* by the slow release from a styrene-acrylate copolymer coating. For comparison, they also studied the inhibitory effect through the direct addition of propyl paraben. *S. cerevisiae* inhibition is slow but is sustained under slow release with the direct addition, an adequate incubation period is required to inhibit the bacteria.

## 5. Conclusions

Food packaging is very important to protect foods from bacterial infection. Therefore, we should develop cheaper, safer food packaging to meet the fast-growing demand of the food industry. According to its inherent characteristics, polymer food packaging can be divided into two types: biodegradable and non-biodegradable. Although non-degradable food packaging is still the most commonly used material, their use is being reduced as a result of their environmental pollution. Therefore, in the long run, biodegradable food packaging is more promising because it is green, renewable, and environmentally friendly. 

As for antibacterial agents, they can also be divided into natural and synthetic materials. Although the toxicity of natural antibacterial agent is low, the production cost is high, which hinders their wide applications. Compared with natural antibacterial agents, Synthetic antibacterial packaging can inhibit the growth of bacteria more effectively. However, the toxicity of synthetic agents should not be ignored. Reducing the toxicity of synthetic material is also an important subject in need of further research. 

In summary, the development trends of antibacterial food packaging materials are as follows:

First of all, new, safe, and highly effective antibacterial agents, such as synthetic antibacterial peptides, should be developed to obtain broad-spectrum activity of packaging materials with high antibacterial activity. Synthetic antimicrobial peptides not only have a good antibacterial activity against various bacteria, but also have the advantages of low cost, high efficiency, and biodegradability [109,110,111]. In comparison with natural antimicrobial peptides, synthetic antimicrobial peptides have more stable properties. Synthetic antimicrobial peptides easily to copolymerize via the ring-opening polymerization of α-amino acid *N*-carboxyanhydrides (NCA) directly with polymers (such as PCL) to form film-forming materials [23,112,113].

Second, the controlled release of antibacterial agents has become an important research direction, enabling the possibility of producing packaging materials with durable antibacterial properties. Introducing micro-nano structures, such as microcapsules and nanofibers, to packing materials helps achieve gradual release and simultaneously maintains the mechanical properties of materials. 

Third, green and environmentally-safe food packaging is a growing trend. Green, environmentally-friendly, biodegradable, low cost, and easy-to-obtain polymers should be ideal candidate substrate materials for food packaging.

With the development of antimicrobial packaging materials, highly comprehensive, systematic and unified standards are needed to evaluate their antimicrobial activity and food safety. Antimicrobial packaging materials should also be provided with smart technology, such as indicators, to show the degree of bacterial infection in food or its microenvironmental factors, such as temperature [114], pH value, and humidity, and to meet high requirements and enrich food packaging applications.

## Figures and Tables

**Figure 1 polymers-11-00560-f001:**
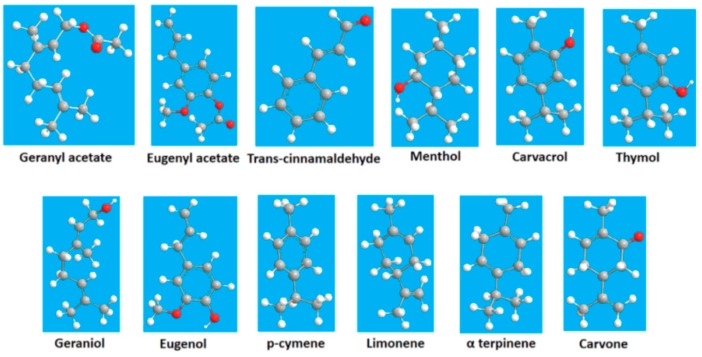
Main chemical components of Eos [40].

**Figure 2 polymers-11-00560-f002:**
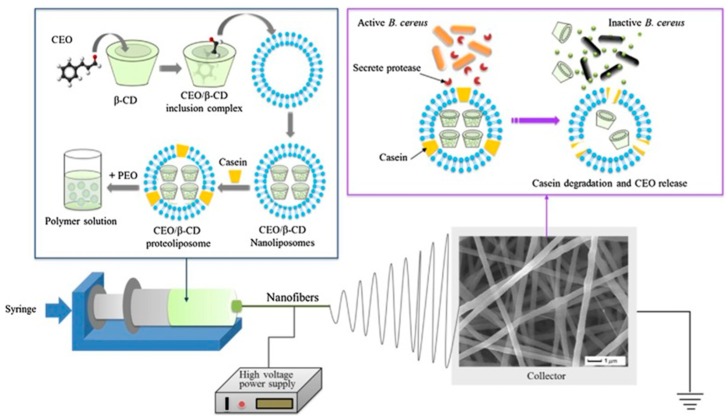
Electrospinning for CEO/β-CD proteoliposomes incorporated into PEO nanofibers. And schematic of B. cereus proteinase-triggered CEO release from CEO/β-CD proteoliposomes [49].

**Figure 3 polymers-11-00560-f003:**
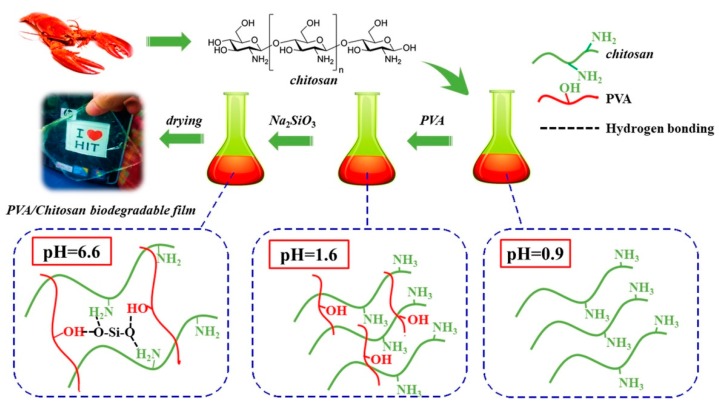
SiO_2_ in situ enhanced PVA/CS biodegradable films by hydrolysis of sodium metasilicate [87].

**Table 1 polymers-11-00560-t001:** Antimicrobial agents incorporated into polymeric matrix and their applications in food packaging.

Antimicrobial Agents	Polymeric Matrix	Foods	Targeted Microorganisms	References
*Origanum vulgare* and *Thymus vulgaris*	LDPE	/	*Salmonella. typhimurium*, *Listeria. monocytogenes*, *Escherichia. coli*	[12]
Cinnamon essential oil	Active paper	Sliced bread	*Rhizopusstolonifer*	[13]
Nisin et al bacteriocins	PE, EVA	/	*Listeria. monocytogenes* V7	[14]
Nisin or pediocin	Corn starch	/	*Listeria. monocytogenes* C. *perfringens*	[15]
Plantaricin BM-1	PE, LDPE, HDPE	/	*Listeria. monocytogenes*	[16]
Lysozyme	PP-g-PAA	/	*Listeria. monocytogenes*	[17]
Sodium benzoate	Poly(butylene adipate-co-terephthalate)/organoclay nanocomposite	/	*Bacillus subtilis* and *Staphylococcus aureus*	[18]
Potassium sorbate and oregano essential oil	Thermoplastic starch, poly(butylene adipate-coterephthalate)	Restructured chicken steaks	*Escherichia. coli*	[19]
Potassium sorbate and vanillin	Chitosan films	Butter cake	mould	[20]
Citric acid and chitosan	Fish gelatin/chitosan composite films	/	*Escherichia. coli*	[21]
Gallic acid grafted chitosan	Gallic acid grafted chitosan films	Agaricus bisporus	/	[22]
Chitosan	Acrylonitrile and acrylamide grafted chitosan	Apple and guava	*Escherichia. coli*, *Staphylococcus aureus* and *Pseudomonas aeruginosa*	[23]
Grape fruit seed extract	Carrageenan	/	*Escherichia. coli*, *Staphylococcus aureus*, *Bacillus cereus* and *Listeria. monocytogenes*	[24]
Grape fruit seed extract	Agar/alginate/collagen hydrogel films	Potatoes	*Listeria. monocytogenes*, *Escherichia. coli*.	[25]
Propyl paraben	Styrene-acrylate copolymers	/	*Saccharomyces cerevisiae*	[26]

(LDPE: low density polyethylene; PE: polyethylene; EVA: ethylene vinyl acetate; HDPE: high density polyethylene; PP-g-PAA: polypropylene modified by polymerization of acrylic acid).

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
