# Peer review of "Polymeric Antimicrobial Food Packaging and Its Applications"

_polymers, 2019, doi:10.3390/polym11030560_

Round 1

Reviewer 1 Report

Nowadays, in the food industry, the conventional techniques for bacteria growth suppression such as thermal treating, sublimation, freezing, sterilization and others are actively being substituted by active polymer packaging. So, this compact comprehensive review of the barrier polymers, as well as the synthetic and natural active agents with the antimicrobial performance, is the valuable impact on food science development.

            In the framework of this manuscript, the combination of stable or biodegradable barrier polymers with inherent antibacterial behavior and biocompatibility (e.g. chitosan, PLA) opens the wide gate for the design of innovative food packaging elaboration. Certainly, the content of this paper falls within the scope of the Journal with appropriate terminology and adequate argumentation displaying a good fitting to the general task of the mini review. The literature cited is quite relevant to this survey and the illustrations were executed in an unambiguous and accurate manner with the clear coherent interpretation. The structure of the submission is logically perfect; English is fairly comprehensible.

Along with clear logical consistency of paper, there are a few remarks that, in spite of their minority character, should be taken into consideration.

- The abstract should be enhanced to reflect more completely the sequences of presentation and reflect the principal sections of submission.

- The finalizing remarks reached by the authors are too laconic and do not open the paradigm of the review. It is worth to compare the natural and synthetic materials as it was done for peptides.

- P 1, L 24: “microorganisms, such as Campylobacter, Salmonella, Yersinia 24 enterocolitica, Escherichia coli, and Listeria monocytogenes, which mainly grow on food surfaces [1,2].” There are some findings where it was shown that microorganisms could grow in the polymer volume (especially in pores, in gels with a great mesh)

- P 10 L321: “yikangic acid”  This is clearly an erroneous term.

Nevertheless, taking into account above minor remarks, I would like to recommend this paper for further Editorial execution and the publication in Polymers. 

Author Response

Responses to Reviewer 1

1. The reviewer’s comment:

The Abstract must be expanded and informs better about the subject of this review.

The authors’ reply:

Thank you for your important comments. Abstract has been revised (please see pages 1 line 12-22 in the revised manuscript).

2. The reviewer’s comment:

Keyword – It should be “biodegradable materials”.

The authors’ reply:

Thank you for your important comments. We have changed the keyword “biodegradable” to “biodegradable materials” (please see pages 1 line 23 in the revised manuscript).

3. The reviewer’s comment:

L. 28/29 –Active packaging extends the shelf life of food products.

The authors’ reply:

Thank you for your important comments. We have changed the sentence “Active packaging can inhibit bacterial spoilage to extend their shelf life” to “Active packaging extends the shelf life of food products” (please see pages 1 line 36 in the revised manuscript).

4. The reviewer’s comment:

L. 36 –Denaturization of what?

The authors’ reply:

Thank you for your important comments. It is a mistake and the sentence is revised (please see pages 2 line 42-43 in the revised manuscript).

5. The reviewer’s comment:

L. 51/52 –Wrong style “of food packaging …food packaging…”.

The authors’ reply:

Thank you for your important comments. We have changed the sentence “The main component of food packaging is synthetic organic food packaging” to “The main antibacterial components in food packaging is synthetic organic materials(please see pages 2 line 58-59 in the revised manuscript).

6. The reviewer’s comment:

Table 1 – The plant and bacteria Latin names must be written in italic. and explanations of the abbreviation should be added in footnote.

The authors’ reply:

Thank you for your important comments. We have corrected them in the table1.

7. The reviewer’s comment:

L.74-It should be “Polyvinyl chloride”.

The authors’ reply:

Thank you for your comments. We have changed the “Polyvinyl Chloride” to “Polyvinyl chloride” (please see pages 4 line 82 in the revised manuscript).

8. The reviewer’s comment:

L 126-Chemical structure of chitosan should be briefly described.

The authors’ reply:

Thank you for your important comments. We have added the sentence “Chitosan is produced from chitin with a linear structure, which is constituted by random deacetylated unit and acetylated unit.” (please see pages 5 line 134-135 in the revised manuscript).

9. The reviewer’s comment:

L. 142 – Full name of GRAS is needed.

The authors’ reply:

Thank you for your important comments. We have added the full name of GRAS- Generally Recognized as Safe (please see pages 6 line 151 in the revised manuscript).

10. The reviewer’s comment:

L. 174-Typhi” not in italic.

The authors’ reply:

Thank you for your comments. We have changed the style of “Typhi” to “Typhi” (please see pages 7 line 184 in the revised manuscript).

11. The reviewer’s comment:

L. 184-Electrospinning should be briefly described/explained.

The authors’ reply:

Thank you for your important comments. We have explained the electrospinning that “Electrospinning is one of the most popular methods to prepare nanofibers. With external electric field, polymer solutions or melting polymers can form nanofibers with diameters in the sub-micron range” (please see pages 7 line 194-196 in the revised manuscript).

12. The reviewer’s comment:

L. 208 –It should be “…monocytogenes …”.

The authors’ reply:

Thank you for your comments. We have changed “Monocytogenes” to “ monocytogenes(please see pages 8 line 220 in the revised manuscript).

13. The reviewer’s comment:

L. 228/229 – Too trivial sentence!

The authors’ reply:

Thank you for your important comments. We have simplified the sentence to “Nisin can inhibit the growth of microbes and keep food fresh even at room temperature” (please see pages 8 line 240 in the revised manuscript).

14. The reviewer’s comment:

L. 260 – Bacterial sources of pediocin?

The authors’ reply:

Thank you for your kindly comments. We have added the bacterial sources of pediocin “Pediocin is another commonly used bacteriocin, which is mainly derived from pediococcus acidilactici.” (please see pages 9 line 273-274 in the revised manuscript)

15. The reviewer’s comment:

L. 281-I suggest “Lysozyme” instead of “Enzyme”.

The authors’ reply:

Thank you for your suggestion. We have changed “Enzyme” to “Lysozyme” (please see pages 9 line 295 in the revised manuscript)

15. The reviewer’s comment:

L. 294-It should be “zein”.

The authors’ reply:

Thank you for your comments. We have changed the words “Zein” with “zein” (please see pages 10 line 308, 311, 312 and 313 in the revised manuscript).

16. The reviewer’s comment:

L. 321-Benzoic acid belongs to phenolic acids, not to organic ones.

The authors’ reply:

Thank you for your important comments. We delete the benzoic acid (please see pages 10 line 336 in the revised manuscript).

17. The reviewer’s comment:

L 359 –Explanation of MIC is needed.

The authors’ reply:

Thank you for your important comments. The explanation of MIC is added. (please see pages 11 line 375-377 in the revised manuscript).

18. The reviewer’s comment:

L. 408 –Chemical composition of GFSE should be completed.

The authors’ reply:

Thank you for your important comments. We have added the chemical composition of GFSE that GFSE contains abundant phenolic compounds such as catechins, epicatechin, gallic acid and procyanidins (please see pages 13 line 425-426 in the revised manuscript).

19. The reviewer’s comment:

Conclusion should be added.

The authors’ reply:

Thank you for your important comments. The conclusion has been revised. (please see pages 15 line 510-523 in the revised manuscript).

20. The reviewer’s comment:

References must be prepared in the style of Polymers and reference 22, 23, 44, 64, 78, 100 – Latin names must be in italic.

The authors’ reply:

Thank you for your comments. We have corrected as was asked (please see pages 16-23 line 609, 612, 667, 720, 763 and 819 in the revised manuscript)

Reviewer 2 Report

It is opinion of the reviewer that this paper before acceptance needs several corrections. My individual comments are listed below.

The Abstract must be expanded and informs better about the subject of this review.

Keyword – It should be “biodegradable materials”.

L. 28/29 – Active packaging extends the shelf life of food products.

L. 36 – Denaturization of what?

L. 51/52 – Wrong style “of food packaging …food packaging…”.

Table 1 – The plant and bacteria Latin names must be written in italic.

Table 1 – Explanations of the abbreviation should be added in footnote.

L.74 – It should be “Polivinyl chloride”.

L 126 – Chemical structure of chitusan should be briefky described.

L. 142 – Full name of GRASS is needed.

L. 174 – “Typhi” not in italic.

L. 184 – Electrospinning should be briefly described/explained.

L. 208 – It should be “…monocytogenes …”.

L. 228/229 – Too trivial sentence!

L. 260 – bacterial sources of pedicin?

L. 281 – I suggest “Lysozyme” instead of “Enzyme”.

L. 294 – It should be “zein”.

L. 321 – Benzoic acid belongs to phenolic acids, not to organic ones.

L 359 – Explenation of MIC is needed.

L. 408 – Chemical composition of GFSE should be completed.

Conclusion should be added.

References must be prepared in the style of Polymers.

Reference 22, 23, 44, 64, 78, 100 – Latin names must be in italic.

Author Response

Responses to Reviewer 2

1. The reviewer’s comment:

The abstract should be enhanced to reflect more completely the sequences of presentation and reflect the principal sections of submission.

The authors’ reply:

Thank you for your important comments. Abstract has been revised .(please see pages 1 line 12-22 in the revised manuscript).

2. The reviewer’s comment:

The finalizing remarks reached by the authors are too laconic and do not open the paradigm of the review. It is worth to compare the natural and synthetic materials as it was done for peptides.

The authors’ reply:

Thank you for your important comments. The conclusion has been revised. (please see pages 15-16 line 510-545 in the revised manuscript).

3. The reviewer’s comment:

microorganisms, such as Campylobacter, Salmonella, Yersinia 24 enterocolitica, Escherichia coli, and Listeria monocytogenes, which mainly grow on food surfaces [1,2].” There are some findings where it was shown that microorganisms could grow in the polymer volume (especially in pores, in gels with a great mesh)

The authors’ reply:

Thank you for your comments. We have added the sentence “ there are also some microorganisms growing in the volume of polymers throughout the pores, gels “(please see pages 1 line 30-32 in the revised manuscript).

4. The reviewer’s comment:

yikangic acid”  This is clearly an erroneous term.

The authors’ reply:

Thank you for your comments. It is a mistake and it has been corrected. (please see pages 10 line 336 in the revised manuscript).

Round 2

Reviewer 2 Report

The authors corrected this paper properly taken under considerations all my comments. Therefore I can accept it now.